# Sources and Applications of Emerging Active Travel Data: A Review of the Literature

Mohammad Anwar Alattar [1,*], Caitlin Cottrill [2] and Mark Beecroft [2]

[1] Geography & Environment, School of Geosciences, University of Aberdeen, Aberdeen AB24 3UF, UK
[2] Centre for Transport Research, School of Engineering, University of Aberdeen, Aberdeen AB24 3UE, UK; c.cottrill@abdn.ac.uk (C.C.); m.beecroft@abdn.ac.uk (M.B.)
[*] Correspondence: m.alattar@hotmail.co.uk

**Abstract:** Active travel (AT) has the potential to integrate with, or in some cases substitute for, trips taken by motorized transportation. In this paper we review relevant research on AT outcomes to address the potential of AT and emerging data sources in supporting the transport paradigm shift toward AT. Our analysis identifies physical, mental, built and physical environmental, monetary, and societal outcomes. Traditional methods used to acquire AT data can be divided into manual methods that require substantial user input and automated methods that can be employed for a lengthier period and are more resilient to inclement weather. Due to the proliferation of information and communication technology, emerging data sources are prevailing and can be grouped into social fitness networks, in-house developed apps, participatory mapping, imagery, bike sharing systems, social media, and other types. We assess the emerging data sources in terms of their applications and potential limitations. Furthermore, we identify developing policies and interventions, the potential of imagery, focusing on non-cycling modes and addressing data biases. Finally, we discuss the challenges of data ownership within emerging AT data and the corresponding directions for future work.

**Keywords:** active travel; emerging data sources; crowdsourced data; cycling; Strava; public participation geographic information system (PPGIS)

## 1. Introduction

Active travel (AT), namely journeys that have been undertaken either entirely or partially using human-powered transportation modes such as walking, cycling, or using a wheelchair, has been the focus of much attention due to its potential for remedying negative impacts of urbanization. Among other benefits, AT helps to meet required physical activity guidelines and reduces traffic congestion and pollution [1,2]. Furthermore, AT induces the uptake of emerging micromobility, a term that describes the use of electrically assessed lightweight vehicles such as e-bikes, e-scooters, e-skateboards, and hoverboards. Micromobility transport modes are less physically taxing with a shorter travel duration, reducing the reliance on conventional vehicles, particularly for short journeys [3,4]. More recently, unlike public transportation, AT has played an instrumental role during the COVID-19 outbreak, favoring the practice of social distancing [5]. However, the emphasis of transport planning in most cities is still car-dominant, with policies such as minimum car parking requirements and gas subsidies aiming to reduce car delays across many urban transport networks [6].

In many cases AT can substitute for a large portion of journeys undertaken by motorized transport. For example, in the UK, a quarter of vehicle journeys are under two miles [7], while vehicle journeys under five kilometers (almost 3 miles) constitute around 50% of the total in Europe [8], and in the US, 75% of all journeys are under two miles [9]. Thus, to change this car-dominant transport paradigm, fostering AT requires well-informed policies and interventions, which have often been stalled by inadequate data available from

traditional data sources. Manual data collection methods (e.g., using clickers to obtain AT user volumes) are laborious [10], with much data underreported [11]. Automated methods such as infrared sensors lack contextual information, such as helmet usage and age and gender of active travelers [12]. Nevertheless, traditional data is useful for the validation of emerging data sources.

The advent and ubiquity of information and communication technology, including smartphones and wearable devices, has allowed for emerging AT data (hereinafter emerging data) ventures. These datasets are considered Big Data, characterized by the three v's: volume (very large), variety (highly complex) and velocity (high growth rate), making them unmanageable through traditional methods [13]. Such unprecedented data also, however, provide new opportunities and challenges to aid the transport paradigm shift toward AT. Compared to traditional AT data, emerging data is much more voluminous, less obstructive, and relatively cheaper to collect. For example, Strava ((https://www.strava.com/ accessed on 11 June 2021), a social fitness network where users can track, share and monitor their physical activities such as cycling and running; more about this type of data source is provided in Section 3.2.1) ridership datasets have a fine spatiotemporal resolution whereas traditional counts are limited both spatially and temporally [14]; and traditional safety incident reports (e.g., police and insurance) are underreported compared to BikeMaps ((https://bikemaps.org/ accessed on 11 June 2021), an online platform where users can voluntarily report concerns about cycling safety, including incidents such as collisions, near misses and hazards; more about this type of data source is provided in Section 3.2.3.) data [11].

In an attempt to review the emerging data and address their potential and limitations, this work surveys the current emerging AT data ecosystem, and builds and expands on previous reviews [15,16] to include additional AT modes and new data sources. This review is by no mean exhaustive; rather, it is an effort to provide a representation of current research trends. This literature review was conducted employing Google Scholar using the following terms: active travel, active transportation, cycling, cycle, bicycle, pedestrian, bike sharing systems, big data, data collection, Strava, crowdsourced, emerging data, and traditional data. Titles and abstracts have been screened for inclusion, with a total of 129 references included. This paper aims to assess the state of knowledge on emerging data. Section 2 introduces potential outcomes of AT, while Section 3.1 provides a brief review of traditional AT data sources and Section 3.2 focuses on emerging data. Section 4 discusses open challenges and research directions. Finally, the conclusion is presented in Section 5.

## 2. Active Travel Outcomes

### 2.1. Physical and Mental Wellbeing Outcomes

Urbanization has transformed cities to obesogenic environments, a term that has been coined to describe environments that induce obesity through promoting a sedentary lifestyle and encouraging an excess calorie intake [17]. This has resulted in the first generation to have a shorter life expectancy than their parents [18].

Flint et al. [19] found that in the UK, AT users had a significantly lower body mass index compared to users of motorized travel modes, suggesting that they are less likely to be obese or suffer from related health conditions. In a study based in Osaka, Japan, Hayashi et al. [20] concluded that the duration of walking to work is positively associated with the reduction of hypertension. In Bogota, Colombia, cycling to school has been linked to an improved physical fitness profile compared to motorized transportation [21].

During the COVID-19 pandemic, AT provided transportation that adhered to the social distancing guidelines implemented to reduce the spread of the outbreak. For example, the bike share systems (BSS) in New York were more resilient during the pandemic than the subway [22]. Similarly, in Scotland BSSs were demonstrated as an alternative to public transportation [23]. Sydney exhibited a surge in the willingness to cycle due to hygiene reasons and recreational exposure [24]. In addition, delivery services increased their reliance on cycling during this period [25].

The physical activity gained from AT has been reported to improve mental wellbeing. Physically active individuals in the UK reported less anxiety-related symptoms or emotional distress [26]. Similarly, in Alameda County, California, Camacho et al. [27] demonstrated that inactive individuals are more likely to develop clinical depression.

However, Gelb and Apparicio [28] observed that in Paris, France, close proximity to traffic can impose wellbeing threats to AT users via noise and air pollution. This may erode the corresponding health benefits. In addition to more serious threats resulting from traffic injuries [29], Stelling-Konczak et al. [30] explored the impacts of cyclists' auditory perception (mobile conversation, music and electric cars' quietness) on their safety. The compensatory behavior (i.e., reducing speed, looking around more frequently) of cyclists was found to counterbalance the risk arising from losing auditory cues such as tire and engine noises. Readers are referred to Mueller [31] for a systematic review of AT health outcomes.

### 2.2. Built and Physical Environmental Outcomes

On a global scale, motorized vehicles are the second largest source of carbon emissions [32]. They also contribute to heat emission, ultimately magnifying the urban heat island effect. This in turns reduces satisfaction with the ambient temperature, also known as thermal comfort [33]. The urban heat island phenomenon occurs when a city experiences a higher temperature than its surrounding areas due to the heat retention and emissions associated with anthropogenic activities [34]. Since AT modes do not require fuel and produce substantially less heat, shifting from motorized transportation to AT will reduce climate change severity and the urban heat island effect [35,36]. Moreover, cities are designed to accommodate motorized transportation through highways, parking and tunnels. Shifting to AT requires less real estate allocated to these travel modes. For example, a car parking space is equivalent to 7–9 bicycle parking spaces [37].

### 2.3. Monetary Outcomes

Motorized transportation is associated with numerous overhead costs, such as fuel, insurance, and maintenance, whereas AT is much cheaper (or even free) [38]. Moreover, the extent to which a neighborhood is walkable (ranging from car-dependent to fully walkable) and cyclable (ranging from somewhat bikeable to fully bikeable) has been found to affect home values. Rauterkus & Miller [39] demonstrated a significant correlation between property values and walk scores when measuring walkability using sample properties from Jefferson County, Alabama, USA. Similarly, Lucchesi et al. [40] determined the effect of walkability in two Brazilian cities, namely São Paulo and Rio de Janeiro, to be positive and significant. Li & Joh [41] determined that bike scores exhibited a positive correlation with transit accessibility and property values in Austin, Texas, USA. These results collectively suggest that AT infrastructure investment can yield higher property values.

The New York City Department of Transportation [42] reported that retail sales increased by 49% when protected bicycle lanes were installed on 8th and 9th Avenues, compared to a 3% increase borough-wide, in Manhattan, New York. The Oakland Department of Transportation [43] also reported economic growth of 9% in retail sales after improvement via the Telegraph Avenue project in Oakland, California between 20th and 29th Streets. In particular, the project introduced eight high-visibility pedestrian crosswalks and bike lanes that stretched for nine blocks with parking protection to prevent vehicles from parking. This finding has been confirmed elsewhere, where sales and footfalls are proportional to AT users, for example in Toronto, Canada [44] and Auckland, New Zealand [45].

### 2.4. Societal Outcomes

Economic and social development may be stalled as a result of inequitable transportation [46]. Additionally, enabling adequate societal participation, known as social inclusion, can be promoted through AT as it provides equitable accessibility and availability (transport

equity). Transport equity encompasses: (i) horizontal equity, where fairness is established between individuals who are in the same class of wealth and ability; and (ii) vertical equity, where fairness is established between different income and social classes [47]. Another desirable societal outcome is social interaction, where people engage in mutual leisure activities such as walking or cycling [48]. This in turn can enhance community livability and add a sense of social cohesion [49].

Indirectly, the social benefit of cycling and walking in the European Union has been estimated as €0.18 and €0.37 per kilometer using these modes, respectively. For automobiles, however, a social cost of €0.11 per kilometer has been estimated. This cost-benefit analysis includes numerous parameters such as environmental impact (cost of climate change impact; air, water and ground pollution; noise and space required for infrastructure), travel time and vehicle operation (cost of ownership and operation of a particular transport mode; travel time; roadway congestion imposed on other users), and other factors such as healthcare system savings, perceived safety and discomfort, and quality of life [50].

## 3. Active Travel Data Sources

### 3.1. Traditional Data Sources

Traditional methods to collect AT data comprise manual and automated approaches. Manual methods require a low level of technology sophistication and are labor intensive, meaning more user input is required [51]. The primary advantage of such methods is that they can collect additional information on AT users, such as helmet usage, gender, travel direction, and mobile phone usage, and can differentiate between AT user types (e.g., cyclists, pedestrians, and skaters). These methods can also be used as ground truth counts to validate other methods [52]. Diogenes [53] found video footage to be the most accurate manual method. Similarly, Ryus et al. [54] reported that video footage adjustments, such as speeding up or slowing down at the collector's convenience, can increase accuracy. However, these methods generally require pre-training for accurate results and are also labor-intensive, time-consuming, monotonous, and subject to vagaries of weather. Thus, data collection is typically limited to short time periods [10].

Automated methods involve more advanced technology compared to manual methods. These methods replace human data collectors, and therefore require less or no user input and can be implemented for lengthier periods of time, irrespective of inclement weather conditions [12]. Data collectors have to pay attention to the mechanisms of these sensors, as these may preclude data from some travel modes being collected. For example, magnetometers are not suitable for non-ferrous metal objects (e.g., pedestrians). However, unlike manual methods, automatic methods cannot provide additional information on AT users. In addition, some sensors are further limited in their detection of AT users. Table 1 provides a summary of traditional methods.

**Table 1.** Summary of traditional methods for generating traditional AT data.

| Method | Description |
|---|---|
| **Manual Methods** | |
| Video recording | A standard video camera mounted and directed (temporarily or permanently) in the path of AT users (sidewalks or multi-use trails). The footage is manually examined by the data collector using paper sheets, a handheld counter, or computer software [55]. |
| Travel survey | Travel surveys ask subjects to describe their travel activities or any further information. Data collection methods are based on a range of instruments, such as GPS devices, interviews, and conventional web-based questionnaires [12]. |
| Handheld counter | The use of handheld counters (also known as clickers or tally counters) to count AT users. The data collector can count up to 4,000 AT users per hour [56]. |
| Ride-along observations | The observant collects data from participants during their trips. For instance, the data collector cycles with a study subject to perform a survey or an interview [57]. |
| **Automated Methods** | |
| Pneumatic tubes | Two rubber tubes are stretched across roadways or pathways, perpendicularly attached to the pavement surface. When a bicycle or wheelchair passes over the tubes, a pulse of air is generated, triggering an electrical conduct that registers a count. The distance between the two tubes is programmed to determine the speed. This sensor is highly consumable, with a lifetime ranging from days to months [52,58]. |
| Infrared sensors | Sensors utilize invisible light to detect AT users. There are two main types of sensors: active and passive. Active infrared instruments count AT users when the beam between the transmitter and the receiver is broken. Passive infrared sensors identify temperature variations as AT users move through the detection zone of the sensor. Note that surface temperatures can affect the accuracy of the sensor [52,58]. |
| Magnetometers | Magnetometers detect changes in magnetic fields within the approximation of the sensor created by ferrous metal objects; thus, this sensor is not suitable for non-ferrous metal objects (e.g., carbon-fiber bicycles, pedestrians). The sensor is battery-powered and can be installed below the cycle path. Data are collected through radio communication [59]. |
| Pressure and acoustic pads | A pressure pad sensor detects changes in weight that occur when AT users step on the detection zone. The sensor is capable of distinguishing between the pressure of cyclists and pedestrians. The acoustic pad sensor is limited to pedestrian counting as it uses ground energy waves caused by feet to detect changes. Both sensors are battery-powered and installed within the ground, making them less prone to vandalism [55,60]. |
| CCTV | CCTV positioned on streets aided by artificial intelligence (AI) is able to generate data counts for pedestrians and cyclists. Cameras take pictures at predefined time intervals, then process those images to count pedestrians and cyclists [61]. |

Although the aforementioned limitations confine the applications of such data, they are typically used to validate, calibrate and in some cases complement emerging data sources. However, traditional data (specifically counts, which are considered to be reliable) often fail to accurately capture the number of AT users. Bunn [62] reported an outlier in Strava bike counter data, resulting from cycling in non-bicycle lanes.

### 3.2. Emerging Data Sources

Emerging methods of AT data collection feature high spatial and temporal coverage due to advances in smart devices [63]. Data obtained from emerging methods differ from traditional approaches, which are often spatially and temporally restricted, labor-intensive, time consuming, and cumbersome [14]. Emerging data can originate from various sources, most of which are considered to be crowdsourced, where a series of users provide data addressing the same topic. Table 2 summarizes the emerging data sources adopted to generate AT data.

**Table 2.** Summary of emerging data sources for generating AT data.

| Data Source | Description |
|---|---|
| Social fitness networks (SFNs) | Applications developed by commercial parties to track, share, and analyze personal activity data with user communities using smartphones and wearable devices. |
| In-house developed apps | Applications developed by agencies/organizations that gather AT user information for their own use. |
| Participatory mapping | Engages ordinary users to contribute with their spatial knowledge both qualitatively and quantitatively through a range of methods. |
| Imagery | Extracts infrastructure features and other relevant data from street view and aerial imagery. |
| Bike sharing systems (BSSs) | Systems that provide rental bicycles for users during a certain period of time and generate related datasets. |
| Social media | Geotagged digital footprints available on various social media platforms provide traces on AT. |
| Other | Other sources that do not fall into any of the aforementioned categories. |

Willberg et al. [57] surveyed the relevant research to evaluate traditional methods (counters and observations), BSSs, GPS tracking, SFNs, surveys and interviews, Public Participation Geographic Information Systems (PPGIS), and other sources in terms of their spatial and temporal patterns, demographics, trip purpose, determinants, and barriers.

3.2.1. Social fitness Networks

The phrase "social fitness" has its origins in physical exercise, weight loss regimes, and means of motivating individuals to achieve their fitness goals. Likewise, SFNs allow users to track and share their various physical activity (e.g., walking, cycling, swimming, handcycling, skiing, etc.) data with online communities [64]. These data are owned and managed by private companies such as Strava, MapMyFitness (https://www.mapmyfitness.com accessed on 11 June 2021), and Fitbit (https://www.fitbit.com accessed on 11 June 2021), which in turn distribute the data commercially [12]. Table 3 presents selected studies on SFN applications. Evidently, Strava is the dominant dataset in this category. Lee and Sener [65] thoroughly review the relevant literature on Strava.

**Table 3.** Selected studies on SFN applications.

| Authors | Remarks | Dataset |
|---|---|---|
| Ferster et al. [66] | Identified cyclist incident hotspots. | Strava |
| Alattar et al. [14] and Orellana and Guerrero [67] | Explored the influence of street network analysis on cyclists' route choices. | Strava |
| Hong et al. [68] | Investigated the role of cycling infrastructure in encouraging individuals to cycle in adverse weather conditions. | Strava |
| Hong et al. [69] | Examined the extent to which cycling infrastructure influenced cycling during the COVID-19 lockdown. | Strava |
| Sub and Mobasheri [70], Sun et al. [71] and Lee and Sener [72] | Assessed AT users' air pollution exposure. | Strava |
| Wang et al. [73] | Examined the impact of social (i.e., social network size), personal, psychological, and environmental/situational factors on physical activity. | Fitbit |

These data sources can at times overrepresent certain demographic segments such as male, younger, and tech-savvy users [74]. In addition, Strava is associated with several privacy issues, such as unintentionally revealing military outpost locations [75]. Recent changes in Strava data specifications to maintain user anonymity have consequently resulted in information loss [76] and the data are also restricted by high data acquisition costs

due to the high fees [77]. Despite these limitations, numerous studies have investigated the technical applications of the data. Strelnikova [78] compared Strava and Endomondo (a SFN that has been retired) in terms of spatial and temporal resolution in South Florida, concluding that although Strava provides more detailed information, Endomondo contains data on small road segments and off-road tracks. Livingston et al. [79] address the challenge of Strava representativeness in Glasgow, UK, by comparing Strava (cycling and pedestrian) counts with city center ground-truth cyclist and pedestrian counts. The results reveal that Strava adequately represents locations popular with cyclists.

### 3.2.2. In-House Developed Apps

In-house developed apps, also known as regional bicycling tracking apps, offer region-wide cycling data through GPS-oriented travel diaries that provide GPS traces, trip purpose and demographic information. These apps are generally developed by or for public agencies and aim to record cycling travel patterns for app users in order to improve cycling within the community [12]. For example, CycleTracks (https://www.sfcta.org/tools-data/tools/cycletracks accessed on 11 June 2021) was initially developed by the San Francisco County Transportation Authority (SFCTA) for San Francisco, California. However, the success led a number of agencies and municipalities (e.g., Austin, Texas; Seattle, Washington; and Salt Lake City, Utah) to adopt the app. Other cities have rebranded the app, including Lane County, Oregon (LaneTracks); Atlanta, Georgia (Cycle Atlanta); and Philadelphia, Pennsylvania (CyclePhilly) [80]. Table 4 reports selected studies on the application of in-house developed apps.

**Table 4.** Selected studies on In-house developed apps.

| Authors | Remarks | Dataset |
|---|---|---|
| Hood et al. [81] | Modeled cyclists' route choice to discover cyclists' favored street attributes. | CycleTracks |
| Griffin and Jiao [82] | Examined the proportion of cyclist volume represented by CycleTracks and Strava. | CycleTracks |
| Dhakal et al. [83] | Assessed wrong-way cycling trips. | CyclePhilly |
| Park and Akar [84] | Examined the factors impacting cyclists' detouring decisions. | CycleTracks |

Although more nuances are provided by in-house developed apps (i.e., disaggregated data at the track level) compared to SFNs (i.e., aggregated data at the street level), participant recruitment is considered the main challenge for deploying in-house developed apps, due to the time-consuming and effort-intensive properties [15].

### 3.2.3. Participatory Mapping

Spatial knowledge from ordinary/non-expert users form datasets that can be collected through Volunteered Geographic Information (VGI), a Public Participation Geographic Information System (PPGIS). VGI platforms (e.g., BikeMaps, BikeLaneUpRising (https://www.bikelaneuprising.com/ accessed on 11 June 2021) and SafeLanes (https://safelanes.org/ accessed on 11 June 2021)) promote user engagement in the form of voluntary reporting of various issues that can implicate transportation planning [12]. PPGIS platforms (i.e., Maptionniare (https://maptionnaire.com/ accessed on 11 June 2021) and KoBo Toolbox (https://www.kobotoolbox.org/ accessed on 11 June 2021)) are map-based surveys that solicit spatial and nonspatial information input by inviting respondents [85]. These platforms are usually operated by researchers or practitioners. Table 5 presents selected studies on PPGIS.

**Table 5.** Selected studies on participatory mapping.

| Authors | Remarks | Dataset |
|---|---|---|
| Moran [86] | Explored bike lane blockages. | SafeLanes |
| Ferster et al. [66] | Identified cyclists' incidents hotspots. | BikeMaps |
| Gerstenberg et al. [87] | Identified hotspots of AT users in urban forests. | Maptionnaire |
| Hologa and Riach [88] | Addressed bike hazards and their relationship to certain lane types. | KoBo Toolbox |

The varied mapping skills and familiarity of study areas among users may result in data inconsistencies for such data sources [85]. In addition, participatory mapping platforms are subject to vandalism through false data entries [89].

### 3.2.4. Imagery

High spatiotemporal resolution imagery obtained at low or no cost from satellites (e.g., Google Maps and Bing Maps), street view sources (e.g., Google Street View), or drones can be integrated into supervised or unsupervised methods to extract stationary (e.g., infrastructure) and non-stationary (e.g., ridership) data. Additionally, based on satellite imagery, volunteers can digitize identifiable features including roads and building footprints via the collaborative mapping project OpenStreetMap (OSM). Accordingly, many studies use OSM to extract street networks, a key factor in AT studies. Table 6 presents selected studies on the use of imagery in AT research.

**Table 6.** Selected studies on imagery.

| Authors | Remarks | Dataset |
|---|---|---|
| Wijnands et al. [90] | Identified safe intersection designs. | OSM |
| Moran [91] | Assessed angled parking and its impact on bike networks. | Google Maps |
| Goel et al. [92] | Assessed travel patterns, including walking and cycling, through auditing road infrastructures. | Google Street View |
| Kim [93] | Used a drone to obtain a dataset that spatiotemporally represents pedestrian and bicycle volume. | Drone |
| Boeing [94] | Analyzed the walkable and drivable street networks of 40 US cities. | OSM |
| Yen et al. [95] | Analyzed the walkable, bikeable and drivable street networks of Phnom Penh, Cambodia | OSM |

### 3.2.5. Bike Sharing Systems

BSSs allow for short-term bike rental with pickup and return locations (docks) across areas denoted as docked BSSs (known as third generation systems). In contrast, dockless BSSs (known as fourth generation systems) allow users to unlock and leave rental bikes within a geofence site [96]. BSS-conducted trips are generally less than 30 min [97] and the systems play a key role in increasing the connectivity between public transport and origin or destination locations (first mile/last mile) [98]. BSSs are considered part of "slow tourism", suggesting that AT engages tourists with the exploration of destinations [99]. Interestingly, many BSS operators provide their data openly, such as Citi Bike (https://www.citibikenyc.com/system-data accessed on 11 June 2021) (New York City), Santander (https://cycling.data.tfl.gov.uk accessed on 11 June 2021) (London, England), Metro Bike Share (https://bikeshare.metro.net/about/data/ accessed on 11 June 2021) (Los Angeles, California), and Just Eat (https://edinburghcyclehire.com/open-data accessed on 11 June 2021) (Edinburgh, Scotland). Table 7 details selected studies on BSSs. In addition, Bike Share

Research (http://www.bikeshare-research.org accessed on 11 June 2021) is a collaborative project that provides an application programming interface (API) to retrieve BSS data from global operators.

**Table 7.** Selected studies on BSSs.

| Authors | Remarks | Dataset |
|---|---|---|
| Teixeira and Lopes [22] | Examined the resilience of the Citi Bike BSS during the COVID-19 pandemic. | Citi Bike |
| El-Assi et al. [100] | Analyzed the impact of built environment and weather on BSS demand. | Bike Share Toronto |
| Wang and Akar [101] | Explored gender difference factors affecting Citi Bike ridership. | Citi Bike |
| McKenzie [102] | Compared spatiotemporal patterns between docked and dockless BSS. | LimeBike&Capital BikeShare |
| Eren and Uz [103] | Reviewed factors impacting BSS demand. | N/A |
| Buning and Lulla [104] | Compared the bike-share usage spatiotemporal patterns of visitors and local residents. | Pacers |

The spatial and temporal bike rebalancing issue is one of the main challenges of BSSs, where certain locations at certain times (e.g., rush hours) suffer from bike shortages causing user dissatisfaction and reducing service reliability [105]. This issue may increase overheads, as operators have to instruct vehicles to reestablish the balance. In order to optimize the way BSSs operate, the provided data have been utilized to further investigate and mitigate this challenge. For example, Singla et al. [106] propose financial incentives for users to pick up or drop off bikes in alternate locations.

The vast majority of BSS dataset records are provided in origin–destination journeys rather than routes. Buning and Lulla's [104] work has, however, incorporated GPS data that reveals information about the used routes rather than just origin and destination. Furthermore, BSS datasets may be detailed enough to infer many useful attributes about the user such as subscription type (annual or casual), gender, year of birth, trip timestamp, and home zip code.

### 3.2.6. Social Media

Social media platforms have great potential as reliable, cost-effective, and timely information sources [107]. Through mining techniques, researchers can extract user perceptions on certain topics, whereby user locations can be inferred from geotags [108]. These data have long been acquired from surveys, which require effort in recruiting the sample and may be hindered by low response rates [109]. Thus, transport policies can harvest information from social media to monitor traffic in real time, model travel behavior and demand, and qualitatively analyze facilities' service qualities [110]. Table 8 shows selected studies on social media.

**Table 8.** Selected studies on social media.

| Authors | Remarks | Dataset |
|---|---|---|
| Bhowmick et al. [111] | Estimated pedestrian traffic using georeferenced tweets. | Twitter |
| Wakamiya et al. [112] | Measured pedestrian congestion using georeferenced tweets. | Twitter |
| Das et al. [113] | Conducted text mining to understand bike commuters' sentiments and motivation. | Twitter |
| Wu et al. [114] | Assessed the usage of social media as proxies for urban trails. | Twitter & Flickr |

Despite their benefits, social media data are subject to age group bias and inconsistencies in the data collection [109].

### 3.2.7. Other

In the following, we detail additional sources that do not fall into the above categories. Using GPS tracking apps (e.g., Gaia GPS (https://www.gaiagps.com/ accessed on 21 June 2021)), subjects can record their trips and donate them to researchers. Heesch and Langdon [115] evaluated the usefulness of this type of app in detecting changes resulting from infrastructure improvement on cycling behavior. The work identified a failure in triangulating GPS data due to insufficient traffic-monitoring devices, which may lead to problematic results. In order to overcome this, the authors suggested complementing GPS data with other data sources.

Data service companies (e.g., StreetLight (https://www.streetlightdata.com/ accessed on 21 June 2021)) can aggregate data from different sources to provide a user-friendly analytic platform. Turner [116] determined a high correlation between StreetLight data and ground-truth cyclist counts.

Several self-developed apps and web-based services aim to facilitate crowdsourced data. BikeCitizens (https://www.bikecitizens.net accessed on 21 June 2021) employs user-recorded trips and experiences after they are anonymized to improve cycling in cities. The Bike Data Project (https://www.bikedataproject.org accessed on 21 June 2021) aims to gather data from multiple platforms to improve cycling safety through the donation of user trips.

In response to the COVID-19 pandemic, Apple Mobility Data (https://covid19.apple.com/mobility accessed on 21 June 2021) reports direction requests (walking, driving, and transit) from the Apple Maps app and compares them to a baseline volume from 13 January 2020. The spatial resolution is confined to a country/region, sub-region, or city, with a daily temporal resolution. Using these data, Oguzoglu [117] was able to infer walking trends in Istanbul during the lockdown.

## 4. Open Challenges and Research Directions

### 4.1. Policies and Interventions

Numerous policies that operate at different scales (society, city, neighborhood, and individual) cater to AT. Table 9 presents an overview of policy types that aim to increase AT. Winters et al. [6] determined that more adequate data collection and methodologies are required to optimally implement these policies. The authors explicitly state the need for data improvement and conducted large-scale studies to evaluate these policies. Given the fine spatiotemporal resolution of crowdsourced data, researchers and practitioners can prioritize locations that require policies and interventions and can also justify their investments by quantifying the policy impact.

**Table 9.** Policies to promote AT.

| Policy Level | Description |
| --- | --- |
| Society | Policies to reduce the appeal of motorized vehicles through speed limit reductions and car parking limits, and to promote public transport to incorporate AT. |
| City | Policies to configure urban design through initiatives such as incorporating mixed land use within walking distance to residential areas, the application of car-free centers, reducing block size, and increasing street connectivity. |
| Neighborhood | Policies on AT infrastructure investments to make AT more convenient, comfortable and safe, by adopting separated paths, cycle tracks and end-of-trip facilities (e.g., bicycle parking, showers, lockers). |
| Individual | Policies targeting behavior change, for example through mass media and other campaigns or by providing financial incentives. |

The ongoing need for evidence-based policies and investment make such a practice an open challenge for future research. For example, AT trends have changed as a result of COVID-19 lockdowns worldwide, requirements to meet recommended physical activity levels, and policies to ensure safe commuting [118]. Nurse and Dunning [119] stated that the COVID-19 pandemic exposed the current street setting vulnerability to accommodate AT. Furthermore, most streets do not maintain the recommended distance between people (2 m). These exceptional circumstances, including the lockdown, travel restrictions, and curfews, demand appropriate polices and interventions to accommodate AT during such conditions, and the embedding of these practices in future transport planning activities.

### 4.2. Imagery

High-definition imagery can potentially be incorporated into AT studies. The employment of free and commercial images allows researchers to obtain data on features known to improve the AT experience, namely green spaces and water bodies [120,121]. These two features can be delineated using multispectral imagery through spectral indices such as the normalized difference vegetation index (NDVI) and normalized difference water index (NDWI), respectively. Researchers and practitioners can adopt the cloud service Google Earth Engine to manage, store and process the large amount of data [122].

### 4.3. Non-Cycling Modes

Studies on cycling and micromobility (refer to [123–125] for micromobility studies in Austria, San Francisco, California, and Austin, Texas respectively) modes are rapidly increasing. This is attributed to the marketing campaigns of SFN apps and the proliferation of share schemes, where vehicles are tracked and monitored by service providers for management and operational benefits. In contrast, products used to track non-cycling modes are limited. Venter et al. [126] incorporated pedestrian activities (running, hiking, and waking) to investigate the recreational use of urban green spaces during the COVID-19 partial lockdown in Oslo, Norway. Griffin et al. [127] claimed that pedestrian studies are limited, while Oliveira et al. [25] indicated that emerging micromobility and conventional non-motored vehicles (i.e., skateboards, scooters, and rollerblades) share common challenges and interests with conventional bikes. Cycling is suitable for journeys that are too long for walking and too short for driving [128]. The gap in the literature on non-cycling modes, including mobility aids for those with less mobility, creates an opportunity for future researchers to conduct more studies on these modes. The development of novel methods or products to collect data from non-cycling modes might help to fill this gap.

### 4.4. Biases

Previous research has indicated the biases of emerging data, which in turn threaten the outcome legitimacy of these data. In particular, Strava is associated with several biases: (i) demographic bias towards young and white males [74]; (ii) social desirability bias, where the recorded trips may over-reflect trips with a sense of achievement and overlook mundane journeys [13]; and (iii) self-selection bias that arises when the participants can include or exclude themselves from the sample [129]. BSSs provide data on all their users, eliminating the potential of social desirability and self-selection biases. Although BBSs tend to be more reflective of casual cyclists and visitors, the data are subject to spatial anonymity as the origin (or destination) represents the check-in (or -out) location of the bicycle [57]. Participatory mapping and in-house developed apps require more effort to achieve higher visibility and increase representativeness. Clearly, the digital divide in emerging data favors technology-competent users. Ground-truth data are required for the correction and validation of the crowdsourced data to achieve the desired outcomes. Combining multiple data sources, also known as data fusion techniques, has great potential in overcoming the data uncertainties and biases.

*4.5. Data Ownership*

The increasing number of studies focusing on BSSs illustrates the merits of open data, whereby the data are openly accessible to the public. Such a practice facilitates replicability and prompts more researchers to attempt to answer questions using these data. Open data may also provide an opportunity for VGI platforms to engage researchers with their platforms as data collection instruments and increase their visibility among used platforms to achieve representative sample sizes.

Since SFN data ownership belongs to third parties, the data are subject to specification changes and acquisition fees that might problematize interpretation, replicability and acquisition, respectively. Raturi et al. [76] stated that, to maintain privacy, the new Strava data specification, which consists of binning counts to the nearest increment of 5 (0, 5, 10. etc.)—meaning a count of 3 becomes 0, 4 is rounded up to 5 and so forth—causes loss of information. Additionally, the acquisition fees are regulated by area size and time span. For example, Strava data for Virginia for the year 2016, which includes 2.5 million activities from 110,000 users, was estimated to be $300,000. Thus, to avoid these constraints, open source apps may to some extent substitute this data source. These challenges should be acknowledged by transport agencies prior to adopting this emerging data source. Otherwise, these agencies can deploy open source in-house developed apps such as [80] to avoid these caveats.

## 5. Conclusions

In this paper we have demonstrated the ability of AT to provide numerous positive outcomes for city dwellers. The gaps in coverage, duration, and granularity of traditional data sources may inhibit the design of effective policies and interventions to encourage AT. However, incorporating new and emerging data sources may help to address these gaps. Given the abundance of emerging data sources and their potential, this work reviews and classifies the relevant data sources into SFNs, in-house developed apps, participatory mapping (both PPGIS and VGI), imagery, BSSs, social media, and other types that do not fit into these categories. Table 10 summarizes emerging data sources, proprietorship, readiness, identified biases, and covered topics.

**Table 10.** Summary of emerging data sources, proprietorship, identified biases and covered topics.

| Data Source | Proprietorship | Readiness | Identified Biases | Topic |
|---|---|---|---|---|
| SFNs | Subject to fees | Ready for analysis | Yes | Ridership |
| In-house developed apps | Open source | Require recruitment | – | Ridership |
| PPGIS | Subject to fees and Open source | Require recruitment | – | Ridership, infrastructure and safety |
| VGI | Open source | Ready for analysis | – | infrastructure and safety |
| Imagery | Subject to fees and Open source | Ready for analysis | – | Infrastructure and safety |
| BSSs | Open source | Ready for analysis | – | Ridership |
| Social media | Open source | Ready for analysis | Yes | Infrastructure and safety |

Researchers, practitioners and data providers should consider the following to magnify AT outcomes:

1.  The impact of policies can be quantified in order to predict the impact of wider-scale transferability;
2.  Imagery can be used to investigate the wide scale (city or region level) impact of water bodies and greenness on AT;

3. Limitations in non-cycling modes can be overcome by further research and newly-developed data platforms, as well as monitoring and tracking products that target these modes;

4. The biases inherent in emerging data allow for the adoption of novel traditional sources—for example, the recent application of drones and CCTV to collect ground-truth data on AT users. Future research can potentially adjust these data using such novel traditional data sources;

5. Transport agencies may consider following the lead of BSS by providing openly accessible ridership, safety, and infrastructure data to allow more research and consequently a better understanding of AT.

**Author Contributions:** Conceptualization, M.A.A., C.C. and M.B.; writing—original draft preparation, M.A.A.; writing—review and editing, C.C. and M.B.; supervision, C.C., M.B. All authors have read and agreed to the published version of the manuscript.

**Funding:** This research received no external funding.

**Institutional Review Board Statement:** Not applicable.

**Informed Consent Statement:** Not applicable.

**Conflicts of Interest:** The authors declare no conflict of interest.

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
