# Peer review of "Sources and Applications of Emerging Active Travel Data: A Review of the Literature"

_sustainability, doi:10.3390/su13137006_

Round 1
Reviewer 1 Report
Dear authors,
Your article is an informative review that provides a comprehensive understanding but also a wide range of options for further reading.
Nonetheless, you may find below some comments and suggestions for your consideration:
- Section 2. In the outcomes, you should further elaborate on the indirect social costs and benefits. For example, these stemming from the improvement in public health, climate - environment and social inclusiveness and the potential to use spare public space due to less need for road infrastructure, in relation to the costs related to road safety.
- Table 1 - Ride-along observations. A brief explanation is needed as to how “The observant collects data from participants during their trips”.
- Table 1 – Automated methods. It is worth mentioning that some of these methods cannot be applied to collect pedestrian data.
- Please check reference on Lines: 172, 235, 251, 266 and 286. The following message is displayed: “Error! Reference source not found”.
- Table 2. While the methods in Table 1 are categorized in terms of technology and technique, the categorization of methods in Table 2 often follows a different approach, mostly referring to the type of service (e.g. a SFN is not a technology/technique for data collection, while magnetometers and surveys are). There should be compatibility in the way traditional and emerging methods are categorized.
- It could be useful to develop an integrated table that correlates the specific considerations/obstacles to data availability and quality with the corresponding methods of Table 2 (see Section 3, e.g. data ownership and acquisition cost for SFN data or socio-demographic biases in Social Media).
- Section 4. The data ownership and the acquisition cost for planning authorities and the research community could be highlighted as a challenge.
Conclusively, I believe that considering the above comments can expand the scientific impact of your paper.
Author Response
Point 1: Section 2. In the outcomes, you should further elaborate on the indirect social costs and benefits. For example, these stemming from the improvement in public health, climate - environment and social inclusiveness and the potential to use spare public space due to less need for road infrastructure, in relation to the costs related to road safety.
Response 1: We added section “2.4. societal outcomes”, where we briefly pointed out social inclusion, transport equity and social cost and benefit that includes the potential benefits of AT in respect to the aforementioned components.
Point 2: Table 1 - Ride-along observations. A brief explanation is needed as to how “The observant collects data from participants during their trips”.
Response 2: We further elaborated this point by providing an example.
Point 3: Table 1 – Automated methods. It is worth mentioning that some of these methods cannot be applied to collect pedestrian data.
Response 3: This has been indicated in lines 238 to 240 and is also included in Table 1.
Point 4: Please check reference on Lines: 172, 235, 251, 266 and 286. The following message is displayed: “Error! Reference source not found”.
Response 4: These errors have been fixed.
Point 5: Table 2. While the methods in Table 1 are categorized in terms of technology and technique, the categorization of methods in Table 2 often follows a different approach, mostly referring to the type of service (e.g. a SFN is not a technology/technique for data collection, while magnetometers and surveys are). There should be compatibility in the way traditional and emerging methods are categorized.
Response 5: We agree and after reviewing previous published literature such as Nelson et al. (2021) and Lee and Sener (2020) we believe the term “emerging data sources” would be more appropriate and consistent with the previous literature.
Point 6: It could be useful to develop an integrated table that correlates the specific considerations/obstacles to data availability and quality with the corresponding methods of Table 2 (see Section 3, e.g. data ownership and acquisition cost for SFN data or socio-demographic biases in Social Media).
Response 6: We agree. Therefore, added Table 10 in Section 5 that compares between emerging data sources in terms of proprietorship, readiness, identified biases and covered topics.
Point 7: Section 4. The data ownership and the acquisition cost for planning authorities and the research community could be highlighted as a challenge.
Response 7: We agree. Subsection 4.5 has been renamed to Data Ownership and we discussed some potential technical issues with data ownership and data acquisition.
Reviewer 2 Report
The authors should review recent papers on bike sharing systems. There are several papers in literature about for bike sharing systems that consider allocation-rebalancing logistic problems. In addition, the authors can classify the literature of bike sharing systems that studies real-cases with big data.
Author Response
Point 1: The authors should review recent papers on bike sharing systems. There are several papers in literature about for bike sharing systems that consider allocation-rebalancing logistic problems. In addition, the authors can classify the literature of bike sharing systems that studies real-cases with big data.
Response 1: We agree that bike sharing systems rebalancing is an essential issue for both service providers and users. Therefore, we highlighted this issue in line 374 to 380. However, we believe classifying literature is more appropriate if our literature review is exclusively focused on bike sharing systems as it might cause inconsistency in providing studies as we do not intend to reclassify other emerging data sources.
Reviewer 3 Report
The subject of the paper is interesting and up-to-date. However, after reading the text carefully, the following points were formulated.
- line 59 "Strava" and line 62 "BikeMaps": merely mentioning a web address in the footnotes does not explain anything about the two commercial systems. The footnotes should be supplemented with a short description of these solutions - especially because it is an Introduction to the paper;
- table 1: Have you not found automatic methods based on video recording and then recognition of objects by methods using deep learning AI?
- line 172: "Error! Reference source not found.";
- line 187: "Error! Reference source not found.";
- line 235: "Error! Reference source not found.";
- line 251: "Error! Reference source not found.";
- line 266: "Error! Reference source not found.";
- line 286: "Error! Reference source not found.";
- there is no "Methodology" section, which is also given more and more frequently in articles on "Literature review" - mainly due to the need to justify the fact that the literature review is sufficiently thorough. Without specifying the methodology, it is difficult to say whether the review is exhaustive and what part of the review is already known in the literature, and which part is the authors' own contribution.
Author Response
Response to Reviewer 3 Comments
Point 1: - line 59 "Strava" and line 62 "BikeMaps": merely mentioning a web address in the footnotes does not explain anything about the two commercial systems. The footnotes should be supplemented with a short description of these solutions - especially because it is an Introduction to the paper;
Response 1: The footnotes have been revised to briefly introduce Strava and BikeMaps.
Point 2: - table 1: Have you not found automatic methods based on video recording and then recognition of objects by methods using deep learning AI?
Response 2: We added CCTV in Table 1 which uses video recording and AI.
Point 3: - line 172: "Error! Reference source not found.";
- line 187: "Error! Reference source not found.";
- line 235: "Error! Reference source not found.";
- line 251: "Error! Reference source not found.";
- line 266: "Error! Reference source not found.";
- line 286: "Error! Reference source not found."
Response 3: These errors have been fixed.
Point 4: - there is no "Methodology" section, which is also given more and more frequently in articles on "Literature review" - mainly due to the need to justify the fact that the literature review is sufficiently thorough. Without specifying the methodology, it is difficult to say whether the review is exhaustive and what part of the review is already known in the literature, and which part is the authors' own contribution.
Response 4: We briefly described our methodology in the last paragraph of the introduction (see line 71 to 77). However, this work is intended to be a representative rather than exhaustive review.
Round 2
Reviewer 3 Report
The authors' additions and explanations are satisfactory.